# Effect of Multiple Annular Plates on Vibration Characteristics of Laminated Submarine-like Structures

**DOI:** 10.3390/ma15186357

**Published:** 2022-09-13

**Authors:** Zhengxiong Chen, Rui Zhong, Shuangwei Hu, Bin Qin, Xing Zhao

**Affiliations:** 1State Key Laboratory of High Performance Complex Manufacturing, Central South University, Changsha 410083, China; 2Key Laboratory of Traffic Safety on Track, Ministry of Education, School of Traffic & Transportation Engineering, Central South University, Changsha 410075, China; 3Joint International Research Laboratory of Key Technology for Rail Traffic Safety, Central South University, Changsha 410075, China; 4National & Local Joint Engineering Research Center of Safety Technology for Rail Vehicle, Central South University, Changsha 410075, China

**Keywords:** Rayleigh–Ritz technology, laminated submarine-like model, annular plate, vibration analysis

## Abstract

A numerical model for the prediction of vibration behaviors of a laminated submarine structure consisting of spherical, cylindrical, and cone shells with multiple built-in annular plates is reported in this article. With the aid of the first-order shear deformation theory (FSDT) concerning plates and shells, the energy expressions of each substructure are derived. The displacement functions in the energy functionals are expanded by the employment of Legendre orthogonal polynomials and circumferential Fourier series. Then, the Rayleigh–Ritz procedure is performed to obtain the eigenfrequency and the corresponding eigenmode of the submarine model. The correctness of the structural model is examined by comparing the results with existing papers and the finite element method, and the maximum deviation is not more than 2.07%. Additionally, the influence of the plate’s thickness, position, inner diameter, as well as the laying angle on the intrinsic vibration characteristics of laminated submarine-like structure is determined. The results reveal that rational geometry design and assemblage benefit the vibration performance of the combination. Increasing the thickness of all the annular plates, decreasing the inner radius, and regulating the laminated scheme, make remarkable influence on structural free vibration, with the maximum relative changing rate of frequency exceeding 97%, 16%, and 23%, respectively.

## 1. Introduction

Submarines, with typical combined conical–cylindrical–spherical structures, play an important role in military reconnaissance, deep-sea exploration, marine scientific research, etc. In terms of specific engineering requirements, such as improving a submarine’s rigidity, vibration suppression, outfitting, and multiple annular plates are always introduced to the coupled structure, which may cause significant changes in the vibration characteristics of the coupled structure. For this reason, a reasonable and clear understanding of vibration characteristics for coupled laminated submarine-like structures considering multiple built-in annular plates is of great significance in terms of their multi-functional design.

On basis of an investigation into the dynamic characteristics of plate and shell structures [1,2,3,4,5,6,7], with the continuous and in-depth research into the vibration mechanism of revolution structures, a series of numerical methods have been developed, including the well-known Ritz technology [8,9,10], the domain decomposition method [11,12], the differential quadrature method [13,14], and other methods [15,16,17]. Parashar et al. [18] used polynomial functions with orthogonality to express the displacement components of a piezoelectric ceramic cylindrical shell and used the Rayleigh–Ritz method to obtain its modal performances. On the basis of the unified Rayleigh–Ritz scheme, Du et al. [8] reported the inherent mechanical properties of a rotating cylindrical shell with a hard coating. Li et al. [9] derived the geometric equations of a conical shell with the help of Reissner’s shell theory, constructed the displacement functions with improved Fourier series, and finally solved the modal features of the conical shell under elastic constraints. To clarify the vibration mechanism of the coupled shell, An et al. [11] used the domain decomposition method to establish a structural analysis model. Choe et al. [12] solved the elastodynamic problems of functionally graded elliptic shells with elastic constraints by using the semi-analytical domain decomposition method. The generalized differential quadrature method was employed by Li et al. [13] to construct a numerical vibration model for a conical shell made of metal foam material under the elastic boundary. Within the framework of the classical plate theory, Saini et al. [14] derived the motion equation of a non-uniform, functionally graded circular plate by using the energy principle and obtained the frequency equation of the circular plate under the classical boundary. Zhang et al. [15] used the dynamic stiffness approach to calculate the vibration properties of cylindrical-conical composite structures with reinforced ribs under multiple classical boundary conditions. Araki et al. [16] studied the free vibration of a cylindrical shell via the Galerkin method, where the displacement functions of the so-called spectral nodes were expanded by Fourier series. Kim et al. [17] analyzed the free vibration of a coupled shell composed of a cylindrical shell, parabolic shell, and hyperbolic shell by using the Haar wavelet discrete method. 

According to the above literature survey, investigations into the vibration behaviors of assembled structures are mostly concerned with numerical algorithms and assembled shell structures. Based on this foundation, the mechanical behaviors of shell–plate coupling structures have been observed. Cao et al. [19] presented an accurate solution to the cylindrical shell–circular plate combination. By means of the modified Fourier series method and considering coupling springs, parametric analyses including the stiffness and conditions of coupling positions and the boundaries were carried out. An improved Fourier series was also applied by Chen et al. [20] to develop a unified theoretical model of an open cylindrical shell–rectangular plate assembled structure with laminated materials in the framework of the first-order shear deformation theory (FSDT). Then, the solutions were obtained by means of the Rayleigh–Ritz method, and comprehensive discussions on the impacts of the main factors on the structural vibration performance were presented, including the geometric properties, coupling positions, and the stiffness of coupling springs. Jin et al. [21] analyzed the vibro-acoustic behaviors of a submarine hull which was symbolled by a conical–cylindrical–hemispherical shell combination and surrounded by heavy fluid. Kim et al. [22] used the Haar wavelet discretization method to investigate the free vibration of conical–cylindrical coupling structures with laminated materials. To reveal the mechanism of structural vibro-acoustic performance, Qu et al. [23] built an immersed submarine model composed of a rigid propeller, a main shaft, two bearings, and an orthogonally stiffened pressure hull. By introducing Fourier series and Chebyshev orthogonal polynomials, the structural displacement and pressure were expressed with those series. The contributions of the stiffness of the ring and the bearing to acoustic behaviors were studied. With experiments and the finite elements method (FEM) simulations, Wang et al. [24] conducted a study about the vibro-acoustic characteristics of a submarine-like system. Xie et al. [25] discussed the main parametric influence on the free and forced vibration of annular plate–cylindrical shell elastically coupled structures with Flügge shell theory [26] and thin-plate theory. Chen et al. [27] studied the vibrational characteristics of stepped cylindrical shell–annular plate coupling structures considering temperature with Chebyshev polynomials and Fourier series. Zhang et al. [28,29] analyzed the vibration characteristics of shell–plate structures. Sobhani et al. [30,31] investigated the vibration behavior of the combination consisting of multiple shells made of composite materials using the Generalized Differential Quadrature method and FSDT. Bagheri et al. [32] analyzed the free vibration of a joined shell structure composed of cylindrical and spherical shells with functionally graded material. Shi et al. [33] studied the vibration of conical-cylindrical shell assembled structure made of functionally graded materials with respective to environment.

The aforementioned papers are almost concerned with shell–shell or shell–plate coupling structures; studies regarding submarine-like structures are scarce. In addition, submarine-like structures made of laminated materials are widely used in reality due to the advantages of laminated materials compared to conventional materials, such as their strength-to-weight ratio and heat and corrosion resistance. Nevertheless, studies regarding the vibration characteristics of submarine-like structures with laminated materials have not been carried out yet. Generally, a submarine-like structure can be treated as a cylindrical shell–spherical shell–conical shell combination, which is divided into several sub-compartments by annular plates according to actual requirements, for example, America’s Ohio-class nuclear submarine and Russia’s Borei-class/Dolgorukiy-class nuclear submarine, etc. Consequently, the unified Rayleigh–Ritz technology in conjunction with the FSDT assumption is applied in the current work to investigate the vibration mechanism of such a combined structure with laminated materials. Furthermore, the effect of the annular plate on the vibration characteristics of the submarine structure is analyzed in detail.

## 2. Analysis Model 

### 2.1. Description of the Model

Figure 1 describes the structural element and coordinate system of a general shell structure. Here, it is assumed that the coordinate system O-αβz is on the middle plane of the shell. Rα and Rβ separately represent the radius of curvature of the shell along α and β directions, when Lα and Lβ are the corresponding lengths of the shell element. h denotes the structural thickness. In addition, artificial spring technology is introduced to simulate boundary conditions. Figure 2 presents a laminated submarine-like model composed of spherical, cylindrical, and conical shells, where J annular plates are arranged. The cone angle of the conical shell is denoted by the symbol α0, and the length of the conical shell along the meridian direction is represented by the symbol Lc. The large end of the conical shell is assembled with a cylindrical shell with length L and radius R1. The right end of the cylindrical shell is connected to the closed hemispherical shell, which means that the radius of the hemispherical shell is consistent with that of the cylindrical shell. The inner radius and thickness of the j-th annular plate are, respectively, expressed by Rj 2 and hj r. The coupling relationship between adjacent substructures is simulated by coupling spring technology.

### 2.2. Kinematic Relations and Stress Resultants

Given that the physical model is provided with geometric features of a medium–thick plate or shell, under the FSDT’s [30] frame, the displacement variables (*U*, *V*, and *W*) of arbitrary points of the kth layer of structures can be written as
(1)$$\begin{array}{l} U(\alpha ,\beta ,z,t)=u_{0}(\alpha ,\beta ,t)+z\xi_{\alpha }(\alpha ,\beta ,t)\\ V(\alpha ,\beta ,z,t)=v_{0}(\alpha ,\beta ,t)+z\xi_{\beta }(\alpha ,\beta ,t)\\ W(\alpha ,\beta ,z,t)=w_{0}(\alpha ,\beta ,t) \end{array}$$
in which the symbols (*u*_0_, *v*_0_, and *w*_0_) are the displacement components of the *k*th midplane. *ξ_α_* and *ξ_β_* are the angular displacement with regard to the *α*-*z* plane and *β-z* plane. The strain components (*ε_α_*, *ε_β_*, *γ_αβ_*, *γ_αz_*, and *γ_βz_*) considered in the *k*th midplane can be written as follows:(2)$$\{\begin{array}{c} \varepsilon_{\alpha }=\varepsilon_{\alpha }^{0}+z\chi_{\alpha },\;\gamma_{\alpha z}=\gamma_{\alpha z}^{0}\\ \varepsilon_{\beta }=\varepsilon_{\beta }^{0}+z\chi_{\beta },\;\gamma_{\beta z}=\gamma_{\beta z}^{0}\\ \gamma_{\alpha \beta }=\gamma_{\alpha \beta }^{0}+z\chi_{\alpha \beta }\;\;\;\; \end{array}$$
where the strains ($\varepsilon_{\alpha }^{0}$, $\varepsilon_{\beta }^{0}$, $\gamma_{\alpha \beta }^{0}$, $\gamma_{\alpha z}^{0}$, $\gamma_{\beta z}^{0}$) on the *k*th midplane, the curvatures (*χ_α_* and *χ_β_*), and the related twist changes (*χ_αβ_*) are stated as
(3)$$\begin{array}{l} \;\varepsilon_{\alpha }^{0}=\frac{1}{A}\frac{\partial u}{\partial \alpha }+\frac{v}{AB}\frac{\partial A}{\partial \beta }+\frac{w}{R_{\alpha }},\;\;\varepsilon_{\beta }^{0}=\frac{\partial v}{\partial \beta }\frac{1}{B}+\frac{\partial B}{\partial \alpha }\frac{u}{AB}+\frac{w}{R_{\beta }},\\ \gamma_{\alpha \beta }^{0}=\frac{B}{A}\frac{\partial }{\partial \alpha }\left(\frac{v}{B}\right)+\frac{A}{B}\frac{\partial }{\partial \beta }\left(\frac{u}{A}\right),\;\;\chi_{\alpha }=\frac{1}{A}\frac{\partial \xi_{\alpha }}{\partial \alpha }+\frac{\xi_{\beta }}{AB}\frac{\partial A}{\partial \beta },\\ \chi_{\beta }=\frac{\partial \xi_{\beta }}{\partial \beta }\frac{1}{B}+\frac{\partial B}{\partial \alpha }\frac{\xi_{\alpha }}{AB},\;\;\chi_{\alpha \beta }=\frac{B}{A}\frac{\partial }{\partial \alpha }\left(\frac{\xi_{\beta }}{B}\right)+\frac{A}{B}\frac{\partial }{\partial \beta }\left(\frac{\xi_{\alpha }}{A}\right),\\ \gamma_{\alpha z}^{0}=-\frac{u}{R_{\alpha }}+\frac{1}{A}\frac{\partial w}{\partial \alpha }+\xi_{\alpha },\;\;\gamma_{\beta z}^{0}=-\frac{v}{R_{\beta }}+\frac{1}{B}\frac{\partial w}{\partial \beta }+\xi_{\beta } \end{array}$$
herein, *A* and *B* represent the Lamé parameters of substructures: (a) cylindrical shell: *α = x*, *β = θ*, *A* = 1, *B* = *R*_1_; (b) conical shell: *α = s*, *β = θ*, *A* = 1, *B* =$s\cdot \sin\alpha_{0}$; (c) spherical shells: *α = φ*, *β = θ*, *A* = *R*_1_, *B* = *R*_1_sin*φ*; (d) annular plate: *α = r*, *β = θ*, *A* = 1, *B* =$s\cdot \sin\alpha_{0}$, *α*_0_ = *π*/2.

According to the generalized Hooke’s law, the relationships between the stresses (*σ_α_*, *σ_β_*, *τ_αβ_*, *τ_αβ_*, and *τ_αβ_*) and strains in terms of linearly elastic materials are given by [34]:(4)$$\left[\begin{array}{c} \sigma_{\alpha }\\ \sigma_{\beta }\\ \tau_{\beta z}\\ \tau_{\alpha z}\\ \tau_{\alpha \beta } \end{array}\right]=\left[\begin{array}{ccccc} {\bar{Q}}_{11}^{k} & {\bar{Q}}_{12}^{k} & {\bar{Q}}_{16}^{k} & 0 & 0\\ {\bar{Q}}_{12}^{k} & {\bar{Q}}_{22}^{k} & {\bar{Q}}_{26}^{k} & 0 & 0\\ {\bar{Q}}_{16}^{k} & {\bar{Q}}_{26}^{k} & {\bar{Q}}_{66}^{k} & 0 & 0\\ 0 & 0 & 0 & {\bar{Q}}_{44}^{k} & {\bar{Q}}_{45}^{k}\\ 0 & 0 & 0 & {\bar{Q}}_{45}^{k} & {\bar{Q}}_{55}^{k} \end{array}\right]\left[\begin{array}{c} \varepsilon_{\alpha }\\ \varepsilon_{\beta }\\ \gamma_{\beta z}\\ \gamma_{\alpha z}\\ \gamma_{\alpha \beta } \end{array}\right]$$
where ${\bar{Q}}_{ij}^{k}\left(i,j=1,2,4,5,6\right)$ represents the elastic stiffness coefficient of the *k*th layer material, which is closely related to Poisson’s ratio and the Young’s modulus of the material. Their expressions are as follows:(5)$$\left[\begin{array}{ccccc} {\bar{Q}}_{11}^{k} & {\bar{Q}}_{12}^{k} & {\bar{Q}}_{16}^{k} & 0 & 0\\ {\bar{Q}}_{12}^{k} & {\bar{Q}}_{22}^{k} & {\bar{Q}}_{26}^{k} & 0 & 0\\ {\bar{Q}}_{16}^{k} & {\bar{Q}}_{26}^{k} & {\bar{Q}}_{66}^{k} & 0 & 0\\ 0 & 0 & 0 & {\bar{Q}}_{44}^{k} & {\bar{Q}}_{45}^{k}\\ 0 & 0 & 0 & {\bar{Q}}_{45}^{k} & {\bar{Q}}_{55}^{k} \end{array}\right]=T\left[\begin{array}{ccccc} Q_{11}^{k} & Q_{12}^{k} & 0 & 0 & 0\\ Q_{12}^{k} & Q_{22}^{k} & 0 & 0 & 0\\ 0 & 0 & Q_{44}^{k} & 0 & 0\\ 0 & 0 & 0 & Q_{55}^{k} & 0\\ 0 & 0 & 0 & 0 & Q_{66}^{k} \end{array}\right]T^{T}$$
(6)$$T=\left[\begin{array}{ccccc} c^{2} & s^{2} & 0 & 0 & -2sc\\ s^{2} & c^{2} & 0 & 0 & 2sc\\ 0 & 0 & c & s & 0\\ 0 & 0 & -s & c & 0\\ sc & -sc & 0 & 0 & c^{2}-s^{2} \end{array}\right],s=\sin\theta_{k},c=\cos\theta_{k}$$
where $\theta_{k}$ represents the fiber-laying angle of the *k*th layer of laminated structures; $Q_{ij}^{k}\left(i,j=1,2,4,5,6\right)$ denote the elastic parameters of materials, which can be obtained by [35]:(7)$$Q_{11}^{k}=\frac{E_{1}^{k}}{1-\mu_{12}^{k}\mu_{21}^{k}},\;Q_{12}^{k}=\mu_{21}^{k}Q_{11}^{k},\;Q_{22}^{k}=\frac{E_{2}^{k}}{1-\mu_{12}^{k}\mu_{21}^{k}},\;Q_{44}^{k}=G_{23}^{k},\;Q_{55}^{k}=G_{13}^{k},\;Q_{66}^{k}=G_{12}^{k}$$

By integrating the stresses along thickness, the force and moment resultant of laminated thick shells can be obtained as:(8)$$N=\left[\begin{array}{ccc} A & B & 0\\ B & D & 0\\ 0 & 0 & A_{s} \end{array}\right]\varepsilon$$
where **A**, **B,** and **D** are the tensile stiffness matrix, coupling stiffness matrix, and bending stiffness matrix, respectively. **A**_s_ is the shear stiffness matrix. The internal element expressions have been reported by Guo et al. [36].

### 2.3. Energy Expressions

The strain energy of the substructure is expressed as follows [10]:(9)$$U_{V}=\frac{1}{2}\int_{S}\left[\begin{array}{l} N_{\alpha }\varepsilon_{\alpha }^{0}+N_{\beta }\varepsilon_{\beta }^{0}+N_{\alpha \beta }\varepsilon_{\alpha \beta }^{0}+M_{\alpha }\gamma_{\alpha }+\\ M_{\beta }\gamma_{\beta }+M_{\alpha \beta }\gamma_{\alpha \beta }+Q_{\alpha }\gamma_{\alpha z}+Q_{\beta }\gamma_{\beta z} \end{array}\right]dS=U_{S}+U_{BC}+U_{B}$$

The strain energy of the structure consists of the potential energy with regard to structure stretching, bending, and stretching–bending coupling. By substituting (3) and (7) into (8), the unified expressions of the strain energy of each substructure can be obtained, of which the expressions are below:(10)$$U_{S}=\frac{1}{2}\iint_{S}\left\{\begin{array}{l} A_{11}{\left(\frac{1}{A}\frac{\partial u_{0}}{\partial \alpha }+\frac{v_{0}}{AB}\frac{\partial A}{\partial \beta }+\frac{w_{0}}{R_{\alpha }}\right)}^{2}+A_{11}{\left(\frac{1}{B}\frac{\partial v_{0}}{\partial \beta }+\frac{u_{0}}{AB}\frac{\partial B}{\partial \alpha }+\frac{w_{0}}{R_{\beta }}\right)}^{2}+\\ \kappa A_{66}{\left(\xi_{\alpha }-\frac{u_{0}}{R_{\alpha }}+\frac{1}{A}\frac{\partial w_{0}}{\partial \alpha }\right)}^{2}+A_{66}{\left(\frac{B}{A}\frac{\partial }{\partial \alpha }\left(\frac{v_{0}}{B}\right)+\frac{A}{B}\frac{\partial }{\partial \beta }\left(\frac{u_{0}}{A}\right)\right)}^{2}+\\ 2A_{12}\left(\frac{1}{A}\frac{\partial u_{0}}{\partial \alpha }+\frac{v_{0}}{AB}\frac{\partial A}{\partial \beta }+\frac{w_{0}}{R_{\alpha }}\right)\left(\frac{1}{B}\frac{\partial v_{0}}{\partial \beta }+\frac{u_{0}}{AB}\frac{\partial B}{\partial \alpha }+\frac{w_{0}}{R_{\beta }}\right)\\ +\kappa A_{66}{\left(\xi_{\beta }-\frac{v_{0}}{R_{\beta }}+\frac{1}{B}\frac{\partial w_{0}}{\partial \beta }\right)}^{2} \end{array}\right\}dS$$
(11)$$U_{BC}=\iint_{S}\left\{\begin{array}{l} B_{11}\left(\frac{1}{A}\frac{\partial u_{0}}{\partial \alpha }+\frac{v_{0}}{AB}\frac{\partial A}{\partial \beta }+\frac{w_{0}}{R_{\alpha }}\right)\left(\frac{1}{A}\frac{\partial \xi_{\alpha }}{\partial \alpha }+\frac{\xi_{\beta }}{AB}\frac{\partial A}{\partial \beta }\right)+\\ B_{11}\left(\frac{1}{B}\frac{\partial v_{0}}{\partial \beta }+\frac{u_{0}}{AB}\frac{\partial B}{\partial \alpha }+\frac{w_{0}}{R_{\beta }}\right)\left(\frac{1}{B}\frac{\partial \xi_{\beta }}{\partial \beta }+\frac{\xi_{\alpha }}{AB}\frac{\partial B}{\partial \alpha }\right)\\ +B_{12}\left(\frac{1}{A}\frac{\partial u_{0}}{\partial \alpha }+\frac{v_{0}}{AB}\frac{\partial A}{\partial \beta }+\frac{w_{0}}{R_{\alpha }}\right)\left(\frac{1}{B}\frac{\partial \xi_{\beta }}{\partial \beta }+\frac{\xi_{\alpha }}{AB}\frac{\partial B}{\partial \alpha }\right)+\\ B_{12}\left(\frac{1}{B}\frac{\partial v_{0}}{\partial \beta }+\frac{u_{0}}{AB}\frac{\partial B}{\partial \alpha }+\frac{w_{0}}{R_{\beta }}\right)\left(\frac{1}{A}\frac{\partial \xi_{\alpha }}{\partial \alpha }+\frac{\xi_{\beta }}{AB}\frac{\partial A}{\partial \beta }\right)+\\ B_{66}\left(\frac{B}{A}\frac{\partial }{\partial \alpha }\left(\frac{v_{0}}{B}\right)+\frac{A}{B}\frac{\partial }{\partial \beta }\left(\frac{u_{0}}{A}\right)\right)\left(\frac{B}{A}\frac{\partial }{\partial \alpha }\left(\frac{\xi_{\beta }}{B}\right)+\frac{A}{B}\frac{\partial }{\partial \beta }\left(\frac{\xi_{\alpha }}{A}\right)\right) \end{array}\right\}dS$$
(12)$$U_{B}=\frac{1}{2}\iint_{S}\left\{\begin{array}{l} D_{11}{\left(\frac{1}{A}\frac{\partial \xi_{\alpha }}{\partial \alpha }+\frac{\xi_{\beta }}{AB}\frac{\partial A}{\partial \beta }\right)}^{2}+D_{11}{\left(\frac{1}{B}\frac{\partial \xi_{\beta }}{\partial \beta }+\frac{\xi_{\alpha }}{AB}\frac{\partial B}{\partial \alpha }\right)}^{2}\\ +2D_{12}\left(\frac{1}{A}\frac{\partial \xi_{\alpha }}{\partial \alpha }+\frac{\xi_{\beta }}{AB}\frac{\partial A}{\partial \beta }\right)\left(\frac{1}{B}\frac{\partial \xi_{\beta }}{\partial \beta }+\frac{\xi_{\alpha }}{AB}\frac{\partial B}{\partial \alpha }\right)\\ +D_{66}{\left(\frac{B}{A}\frac{\partial }{\partial \alpha }\left(\frac{\xi_{\beta }}{B}\right)+\frac{A}{B}\frac{\partial }{\partial \beta }\left(\frac{\xi_{\alpha }}{A}\right)\right)}^{2} \end{array}\right\}dS$$

Accordingly, the kinetic energy of the structure can be expressed as:(13)$$T=\frac{1}{2}\iint_{S}\left\{\begin{array}{l} I_{0}\left[{\left(\frac{\partial u_{0}}{\partial t}\right)}^{2}+{\left(\frac{\partial v_{0}}{\partial t}\right)}^{2}+{\left(\frac{\partial w_{0}}{\partial t}\right)}^{2}\right]+\\ 2I_{1}\left(\frac{\partial u_{0}}{\partial t}\frac{\partial \xi_{\alpha }}{\partial t}+\frac{\partial v_{0}}{\partial t}\frac{\partial \xi_{\beta }}{\partial t}\right)+I_{2}\left[{\left(\frac{\partial \xi_{\alpha }}{\partial t}\right)}^{2}+{\left(\frac{\partial \xi_{\beta }}{\partial t}\right)}^{2}\right] \end{array}\right\}dS$$
where
(14)$$(I_{0},I_{1},I_{2})=\int_{-h/2}^{h/2}\rho \left(1,z,z^{2}\right)dz$$

As stated earlier, the artificial spring technique is introduced to simulate boundary conditions and coupling relationships. Thus, the elastic potential energy stored in the boundary springs is expressed as follows:(15)$$U_{sp}=\frac{1}{2}\int_{-\frac{h}{2}}^{\frac{h}{2}}\int_{0}^{2\pi }\left\{\begin{array}{l} {\left[k_{u0}u_{0}^{2}+k_{v0}v_{0}^{2}+k_{w0}w_{0}^{2}+k_{\varphi 0}\xi_{\alpha }^{2}+k_{\theta 0}\xi_{\beta }^{2}\right]}_{\alpha =0}\\ {\left[k_{u1}u_{0}^{2}+k_{v1}v_{0}^{2}+k_{w1}w_{0}^{2}+k_{\varphi 1}\xi_{\alpha }^{2}+k_{\theta 1}\xi_{\beta }^{2}\right]}_{\alpha =L_{\alpha }} \end{array}\right\}Bd\theta dz$$

The coupling potential energy due to coupling springs can be written as:(16)$$U_{cp}^{i}=\frac{1}{2}\int_{-h_{s}/2}^{h_{s}/2}\int_{0}^{\phi }\left\{\begin{array}{l} k_{u}{\left(u_{0}^{i}-u_{0}^{i+1}\right)}^{2}+k_{v}{\left(v_{0}^{i}-v_{0}^{i+1}\right)}^{2}+k_{w}{\left(w_{0}^{i}-w_{0}^{i+1}\right)}^{2}\\ +k_{\varphi }{\left(\xi_{x}^{i}-\xi_{r}^{i+1}\right)}^{2}+k_{\theta }{\left(\xi_{\theta }^{i}-\xi_{\theta }^{i+1}\right)}^{2} \end{array}\right\}R_{s}d\theta_{s}dz_{s}$$
(17)$$U_{scp}^{i}=\frac{h_{r}^{j}}{2}\int_{0}^{\phi }\left\{\begin{array}{l} k_{u}{\left(u_{0}^{j}-u_{0}^{c,j}\right)}^{2}+k_{v}{\left(v_{0}^{2}-v_{0}^{c,j}\right)}^{2}+k_{w}{\left(w_{0}^{2}-w_{0}^{c,j}\right)}^{2}\\ +k_{\varphi }{\left(\xi_{x}^{2}-\xi_{r}^{c,j}\right)}^{2}+k_{\theta }{\left(\xi_{\theta }^{2}-\xi_{\theta }^{c,j}\right)}^{2} \end{array}\right\}R_{s}$$

As for the overall substructure, the spring potential energy can be expressed as:(18)$$U_{BC}=U_{sp}+\sum_{i=1}^{2}U_{cp}^{i}+\sum_{j=1}^{J}\left(U_{scp}^{j}+U_{cp}^{j}\right)$$

### 2.4. Displacement Admissible Functions and Solution Process

Appropriate displacement admissible functions play an important role in the accuracy of the vibration solutions. Here, the displacement components of the structure in the circumferential direction are expressed as trigonometric series expansions. Additionally, the Legendre polynomials are introduced to ensure and accelerate the convergence of the solution. Thus, the displacement variables are stated as
(19)$$\begin{array}{l} u\left(\alpha ,\beta ,t\right)=\sum_{m=0}^{M}\sum_{n=0}^{N}T_{m}\left(\alpha \right)\left[\cos\left(n\beta \right){\bar{u}}_{mn}\left(t\right)+\sin\left(n\beta \right){\bar{u}}_{mn}\left(t\right)\right]=U\left(\alpha ,\beta \right)u\left(t\right)\\ v\left(\alpha ,\beta ,t\right)=\sum_{m=0}^{M}\sum_{n=0}^{N}T_{m}\left(\alpha \right)\left[\sin\left(n\beta \right){\bar{v}}_{mn}\left(t\right)+\cos\left(n\beta \right){\bar{v}}_{mn}\left(t\right)\right]=V\left(\alpha ,\beta \right)v\left(t\right)\\ w\left(\alpha ,\beta ,t\right)=\sum_{m=0}^{M}\sum_{n=0}^{N}T_{m}\left(\alpha \right)\left[\cos\left(n\beta \right){\bar{w}}_{mn}\left(t\right)+\sin\left(n\beta \right){\bar{w}}_{mn}\left(t\right)\right]=W\left(\alpha ,\beta \right)w\left(t\right)\\ \xi_{\alpha }\left(\alpha ,\beta ,t\right)=\sum_{m=0}^{M}\sum_{n=0}^{N}T_{m}\left(\alpha \right)\left[\cos\left(n\beta \right){\bar{\xi }}_{\alpha ,mn}\left(t\right)+\sin\left(n\beta \right){\bar{\xi }}_{\alpha ,mn}\left(t\right)\right]=\Psi_{\alpha }\left(\alpha ,\beta \right)\xi_{\alpha }\left(t\right)\\ \xi_{\theta }\left(\alpha ,\beta ,t\right)=\sum_{m=0}^{M}\sum_{n=0}^{N}T_{m}\left(\alpha \right)\left[\sin\left(n\beta \right){\bar{\xi }}_{\beta ,mn}\left(t\right)+\cos\left(n\beta \right){\bar{\xi }}_{\beta ,mn}\left(t\right)\right]=\Psi_{\beta }\left(\alpha ,\beta \right)\xi_{\beta }\left(t\right) \end{array}$$
where *N* represents the maximum value of the calculated wave number *n*; *T_m_* is the *m*-th polynomial; *M* is the highest order of the polynomial, also known as truncated values of axial/radial displacement expansions; **U**, **V**, **W**, **Ψ***_α_*, and **Ψ***_β_* represent the function vectors; **u**, **v**, **w**, **ξ***_α_*, and **ξ***_β_* are the coordinate vectors composed of unknown coefficients (${\bar{u}}_{mn}$,${\bar{v}}_{mn}$,${\bar{w}}_{mn}$, ${\bar{\xi }}_{\alpha ,mn}$,${\bar{\xi }}_{\beta ,mn}$).

The employed Legendre orthogonal polynomials are as follows [37]:(20)$$\begin{array}{c} T_{0}\left(\alpha \right)=1,T_{1}\left(\alpha \right)=\alpha \\ \left(p+1\right)T_{p+1}\left(\alpha \right)=\left(2p+1\right)\alpha T_{p}\left(\alpha \right)-pT_{p-1}\left(\alpha \right),\;\;p\geq 2,\;\alpha \in \left[-1,1\right] \end{array}$$

The energy functional of the laminated submarine-like structure is:(21)$$L=\left(T_{S}+T_{C}+T_{L}+\sum_{j=1}^{J}T_{r}^{j}\right)-\left(U_{S,S}+U_{S,C}+U_{S,L}+\sum_{j=1}^{J}U_{r}^{j}\right)-U_{BC}$$

Substituting Equation (18) into Equation (20) and performing a partial derivative operation on unknown coefficients, the vibration equation of the overall model is illustrated as
(22)$$\left(K-\omega^{2}M\right)\vartheta =0$$
where **K** is the stiffness matrix and **M** is the mass matrix. ϑ is the global coordinate vector containing the unknown expansion coefficients of all substructures. By solving the eigenvalues and eigenvectors in (22), the vibration characteristics of the laminated submarine-like model can be obtained.

## 3. Numerical Calculation and Analysis

The free vibration characteristics of the assembled laminated submarine-like structure are further studied based on the established vibration model. To simplify the presentation, three annular plates (i.e., *J* = 3) are used in the follow-up analysis. The geometric parameters of the assembled structure are given as: *L* = 6m, *L*_s_ = 1.5m, *α* = 30°, *R_i_* = 1 m, *r_i_* = 0.4 m, *h* = *hj r* = 0.05 m, *L*_1_ = 0, *L*_2_ = *L*/2, and *L*_3_ = *L*, in which *L_i_* represents the position of the *i*th annular plate in the axial directions of cylindrical shell. If there is no other explanation, all parameters remain as their initial value. In this article, the dynamic characteristics of structure subject to different classical boundary constraints are analyzed. Here, the spring stiffness related to the classical boundary is predefined as: free boundary (F): *k_u_* = 0, *k_v_* = 0, *k_w_* = 0, *k_α_* = 0, *k_β_* = 0; simply supported boundary (S): *k_u_* = 10^14^, *k_v_* = 10^14^, *k_w_* = 10^14^, *k_α_* = 0, *k_β_* = 10^14^; shear–diaphragm (SD): *k_u_* = 0, *k_v_* = 10^14^, *k_w_* = 10^14^, *k_α_* = 0, *k_β_* = 0; clamped boundary (C): *k_u_* = *k_v_* = *k_w_* = *k_α_* = *k_β_* = 10^14^. For the purpose of ensuring the convergence and solution accuracy of the developed model, without the loss of generality, here, the truncation values are configured as: *M*_2_ = 35 for the cylindrical shell, *M*_1_ = *M*_3_ = 20 for spherical and cone shells, and *M_j_* = 18 for all annular plates.

### 3.1. Numerical Verifications

In order to test the correctness and accuracy of the current method, several numerical examples are carried out in the following works. Table 1 exhibits the comparisons between results with the current method and FEM with regard to the laminated cylindrical shell, spherical shell, and conical shell. The relative parameters are: *L*/*R* = 5, *h*/*R* = 0.05, [0°/90°/0°] for laminated schemes, and *E*_2_ = 10.6 GPa, *E*_1_ = 138 GPa, *G*_12_ = 6 GPa, *G*_13_ = G_23_ = 3.9 GPa *μ*_12_ = 0.28, and *ρ* = 1500 kg/m^3^ for the spherical shell; [0°/90°] for laminated schemes, and *E*_2_ = 10 GPa, *E*_1_ = 15 *E*_2_, *G*_12_ = 0.6 *E*_2_, *G*_13_ = 0.6 *E*_2_, *G*_23_ = 0.5 *E*_2_, *μ*_12_ = 0.25, and *ρ* = 1500 kg/m^3^ for cylindrical and conical shells; *R*_0_ = 1 m, *L* = 3 m, and *h* = 0.05 m for the cylindrical shell, and *R*_s_ = 1 m, *L*_S_ = 3 m, *h* = 0.05 m, *α* = 30°. It is noted that the ABAQUS simulation models herein are divided into S4R elements of a 0.04 m × 0.04 m approximate global size. From Table 1, it is visible that results obtained with various methods are in good consistence with each other, demonstrating the correctness and accuracy of this method.

Table 2 shows the natural frequencies of assembled structures corresponding to multiple circumferential wave numbers (i.e., *n* = 1, 2, 3, 4) under classical conditions. The material parameters of the coupling structure made of steel are as follows: *E*_1_ = *E*_2_ = 206 GPa, *ρ* = 7800 kg/m^3^, and *μ*_12_ = *μ*_21_ = 0.3. The results are compared with those from the ABAQUS simulation model divided into 24454 S4R elements of a 0.04 m × 0.04 m approximate global size, of which the overall and axial section views are shown as Figure 3. Here, the abbreviated symbols (namely, C, S, and F) only represent the boundary constraints at the small-diameter edge of the cone shell, and the inner diameter of all plates are unconstrainted. As can be seen from Table 2, the maximum deviation between the frequency results computed by the two numerical methods is 2.07%, which reveals that the presented model can be effectively applied to analyze the frequency characteristics of the coupled laminated submarine-like structure within an acceptable error boundary. On the other hand, the mode shapes corresponding to the frequency results considering the rigidly clamped boundary in Table 2 are depicted in Figure 4. Note that, in view of the fact that annular plates are arranged inside the structure, to facilitate the observation of the mode deformation located on the annular plates, the modal shapes are given in the form of a half-section view. Clearly, the modal shapes obtained by the two methods are highly consistent with each other. For further validation, Figure 5 is shown to compare results obtained by the current method and FEM, taking laminate materials into account, of which the material properties are: *E*_2_ = 10 GPa, *E*_1_ = 15 *E*_2_, *G*_12_ = 0.6 *E*_2_, *G*_13_ = 0.6 *E*_2_, *G*_23_ = 0.5 *E*_2_, *μ*_12_= 0.25, and *ρ* = 1500 kg/m^3^, and the laminated scheme is [0°/90°]. From Figure 5, it is apparent that the numerical results from the two methods share good agreement with each other.

### 3.2. Parametric Study

In what follows, the validated analysis model is employed in this section to analyze the effect of geometric parameters associated with the annular plates on the internal characteristics of the overall laminated submarine-like structure.

First, let us focus on the influence of the position and thickness of the annular plate in the middle of the coupling structure on the inherent characteristics of the coupling structure. Note that only the position of the middle annular plate is changed, whereas the positions of the plates at the first and last ends of the cylindrical shell remain unchanged. Figure 6 shows the variation in the first frequency values, which vary with plate’s geometric parameters, and four constraints are included, with circumferential wave numbers of *n* = 1. It is worth noting that *L*_1_ varying from 1 to 5 with a calculation step of 0.2 is the distance between the middle annular plate and the first fixed annular plate located at the first end of shell. *h*_2_ is the thickness changing with 0.005 increments of the intermediate round plate, which is limited at the interval of [0.05, 0.15]. The material properties are as shown in Figure 5, and the laminated scheme is [0°/90°/0°/90°]. In order to facilitate the analysis, the first mode shapes with *L*_1_ = 5 m and *h*_2_ = 0.15 m are exhibited in Figure 7.

From Figure 6, no matter what the boundary conditions, the natural frequency of the structure will increase with the thickness of the annular plate. This may be explained by the fact that the growth in the thickness of the annular plate contributes to the improvement of the overall rigidity of the structure in the form of reinforcing ribs, which can increase the associated frequencies. Similarly, changes in the position of the annular plate will also have an important impact on the overall rigidity of the structure. For all the boundary conditions herein, when the second annular plate is located in the middle (*L*_1_ = 3 m), the frequency of the structure is the largest, with an increment of over 72% compared to those of *L*_1_ = 1 m or 5 m. When the annular plate moves to both ends, the frequency result decreases. In addition, one phenomenon can be easily observed that the value and trend of the first frequencies are similar for those in Figure 6a–d; the reason may be that the assembled structure owes strong structural stiffness to the coupling relationships between the substructures for one thing; for another, the mode shapes which are presented in Figure 6 and correspond to the first structural frequency are located in the cylindrical shell, of which both ends are combined to other substructures, resulting in the first structural frequency being insensitive to boundaries imposed on the conical shell.

Next, we study the comprehensive influence of the thickness and inner diameter of the three annular plates on the vibration characteristics of the structure in Figure 8. The circumferential wave number is set as *n* = 2, the material and frequency orders discussed here are consistent with Figure 6, the laminated scheme is [0°/90°/0°], and the boundaries are clamped and free boundaries. The thickness of the annular plates varies from 0.16 m to 0.01m with the change step being 0.01 m. The variation range of the inner diameter of the ring plate is [0.3, 0.6], and the step distance is set as 0.02 m. It can be clearly seen from Figure 8 that, for the same modal order, a similar frequency change trend is observed under the free and clamped conditions. In other words, as the annular plates become thicker and the inner diameter decreases, the natural frequency always continues to increase, and the maximum change rates exceed 97% and 16%. Moreover, the first frequency is similar to the same thickness of the annular plate and inner diameter, regardless of the impact of boundaries; the explanation for this is revealed in the discussion about Figure 6.

Finally, the influence of the laying angle on the first eight natural frequency is discussed in detail. As shown in Figure 9, the laminated scheme of the overall structure is [0°/*θ*°/0°], of which *θ* varies from −90 to 90 with an equal step of 10. The circumferential wave number is set as *n* = 2, and the other geometry and material parameters are as shown in Figure 8. From Figure 9, it is apparent that the trend of structural frequency is symmetrical at about *θ* = 0; in the case of *θ* being below zero, the frequency increases over 23% with decreasing *θ* to about −45 and then decreases as *θ* varies from −45 to −90. Similarly, this trend appears as *θ* is over zero.

## 4. Conclusions

This article investigated the vibration performances of a laminated submarine-like structure assembled with multiple built-in annular plates. In view of the well-known FSDT of a shell, the unified energy expressions of structural components, including laminated cylindrical, conical, and spherical shells, as well as annular plates, were deduced. By means of the superposition principle, the energy expression of the assembled structure considering various circular plates was obtained. Finally, the vibration equation was obtained by implementing the Rayleigh–Ritz method. Verification analysis was carried out in numerical examples to present the correctness of the method. Furthermore, the influence mechanism of the annular plate on the dynamic characteristics of the assembled laminated submarine-like structure was investigated by placing the research variables on the geometric parameters of the annular plate. Thus, the following conclusions can be drawn:(1)The established model has good predictive ability regarding the vibration characteristics of laminated submarine-like structures combined with multiple annular plates; the maximum deviation of submarine-like structures is only 2.07%.(2)The influence of the plate’s position on the inherent mechanical properties of the structure is closely related to the modal order, and the frequency of the structure is the largest when the annular plate is set in the middle of the cylindrical shell.(3)The distribution of the laying angle is symmetric to about zero; structural frequencies increase first and then decrease with the increment of the laying angle to some extent.(4)Boundary constraints imposed on the conical shell have little influence on structural vibration as mode shapes relate to the cylindrical shell.(5)The reasonable design of the geometric parameters of the annular plate can effectively improve the rigidity of the structure. Increasing the thickness of all the annular plates, decreasing the inner radius, and regulating the laminated scheme have a remarkable influence on structural free vibration, and the maximum relative changing rates of frequency exceed 97%, 16%, and 23%, respectively.

## Figures and Tables

**Figure 1 materials-15-06357-f001:**
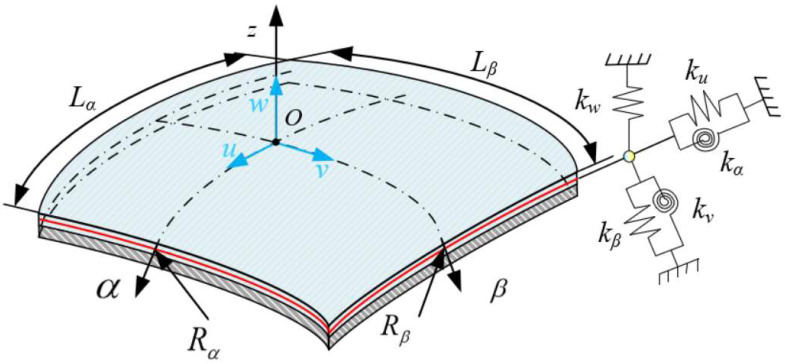
Diagram of shell element coordinates.

**Figure 2 materials-15-06357-f002:**
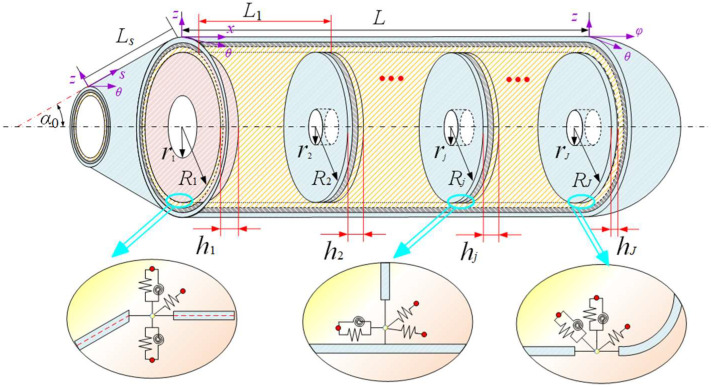
Model diagram of the laminated submarine-like structure.

**Figure 3 materials-15-06357-f003:**
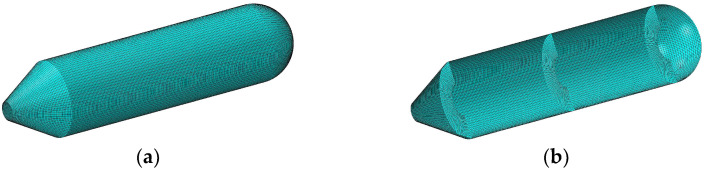
FEM model of submarine-like structure. (**a**) Overall view of FEM model of submarine-like structure; (**b**) Axial section view of FEM model of submarine-like structure.

**Figure 4 materials-15-06357-f004:**
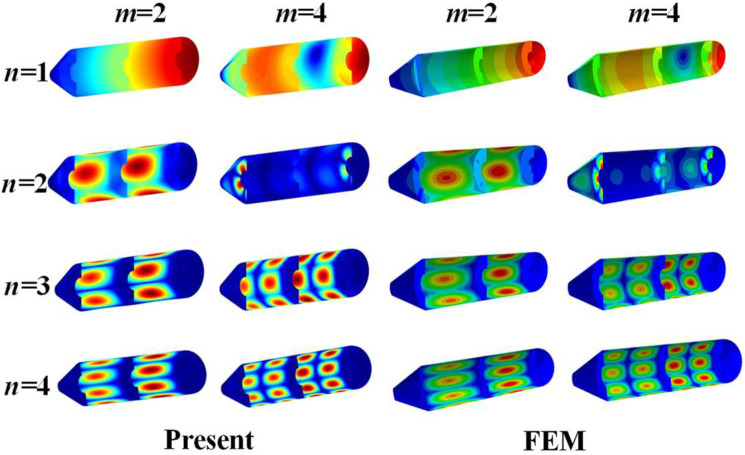
Comparison of modal shapes obtained using the two methods.

**Figure 5 materials-15-06357-f005:**
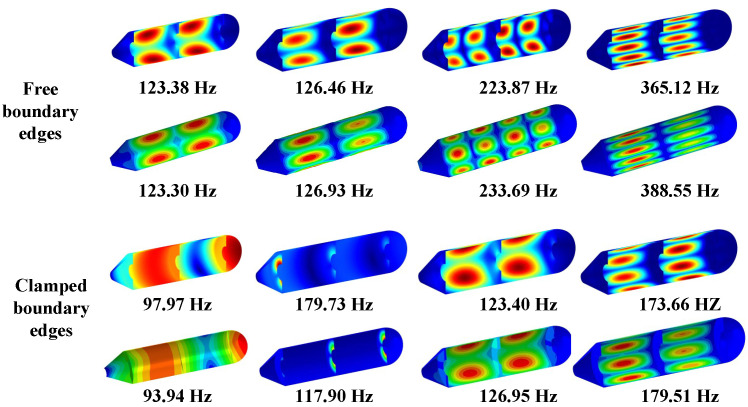
Comparison of modal shapes and corresponding frequencies obtained by the two methods.

**Figure 6 materials-15-06357-f006:**
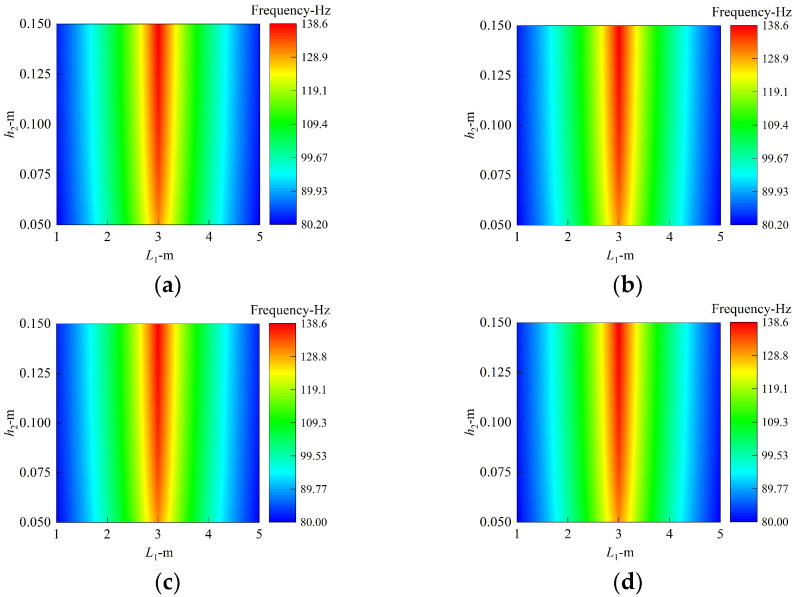
The first frequency value with respect to thickness and position of middle annular plate. (**a**) Clamped boundary. (**b**) Simply supported boundary. (**c**) Shear–diaphragm boundary. (**d**) Free boundary.

**Figure 7 materials-15-06357-f007:**
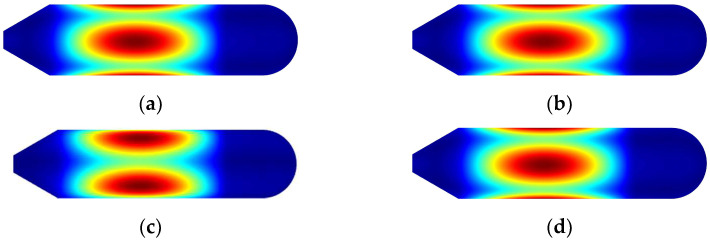
The first mode shapes with L_1_ = 5 m and h_2_ = 0.15 m. (**a**) Clamped boundary. (**b**) Simply supported boundary. (**c**) Shear–diaphragm boundary. (**d**) Free boundary.

**Figure 8 materials-15-06357-f008:**
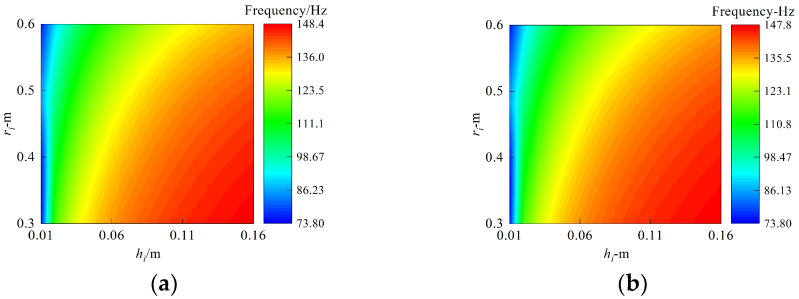
Comprehensive influence of thickness and inner diameter of annular plates. (**a**) Clamped boundary. (**b**) Free boundary.

**Figure 9 materials-15-06357-f009:**
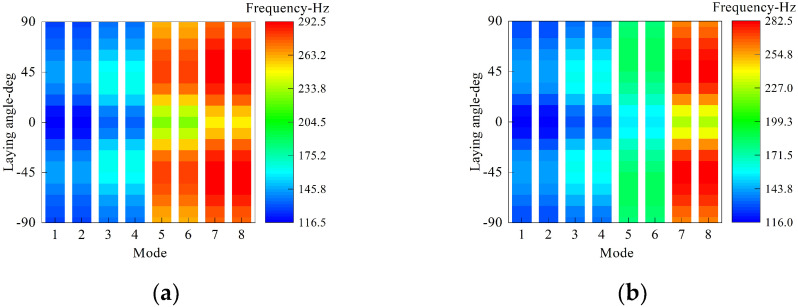
The first eight frequency values with respect to thickness and laminated scheme of the middle annular plate. (**a**) Clamped boundary. (**b**) Free boundary.

**Table 1 materials-15-06357-t001:** Comparison of calculation frequency of spherical, conical, and cylindrical shell with laminated material.

BC	Method	Mode Number							
		1	2	3	4	5	6	7	8
Spherical shell									
F	Present	419.71	508.58	522.89	568.42	603.55	614.46	635.78	670.55
	FEM	417.43	507.55	528.50	557.09	606.15	613.24	637.32	676.48
Conical shell									
F	Present	21.20	53.97	59.74	95.55	135.80	145.74	197.64	204.71
	FEM	21.98	55.83	59.69	62.99	137.69	149.02	194.30	203.63
Cylindrical shell									
C-C	Present	134.31	150.77	176.98	231.98	232.14	237.70	259.37	294.64
	FEM	134.23	150.74	177.04	231.97	232.05	237.47	259.80	294.65
F-F	Present	28.81	31.98	80.96	84.80	154.17	158.21	195.66	206.97
	FEM	28.82	30.90	81.04	84.30	154.42	158.09	195.24	206.44

**Table 2 materials-15-06357-t002:** Comparison of calculation frequency of laminated submarine-like structure under various boundary constraints.

*n*	*m*	C			S			F		
		Present	FEM	Deviation (%)	Present	FEM	Deviation (%)	Present	FEM	Deviation (%)
1	1	6.92	6.78	2.07	6.68	6.54	2.05	143.66	143.56	0.07
	2	94.52	94.46	0.06	94.17	94.11	0.06	213.96	214.28	0.15
	3	213.19	213.46	0.12	213.10	213.36	0.12	215.22	215.48	0.12
	4	214.37	214.67	0.14	214.37	214.67	0.14	304.28	304.32	0.01
2	1	158.15	157.92	0.15	158.15	157.92	0.15	157.93	157.68	0.16
	2	165.67	164.44	0.75	165.67	164.44	0.75	165.42	164.18	0.76
	3	323.77	322.63	0.35	320.16	320.06	0.03	256.37	256.40	0.01
	4	345.01	345.01	0.00	343.93	343.84	0.03	337.88	336.27	0.48
3	1	141.73	141.27	0.33	141.73	141.27	0.32	141.72	141.26	0.32
	2	159.32	158.62	0.44	159.32	158.61	0.45	159.31	158.60	0.45
	3	294.35	293.89	0.16	294.34	293.88	0.16	294.33	293.87	0.16
	4	300.91	298.65	0.76	300.90	298.65	0.75	300.89	298.65	0.75
4	1	201.39	201.38	0.00	201.39	201.38	0.00	201.39	201.38	0.00
	2	209.32	209.28	0.02	209.32	209.28	0.02	209.32	209.28	0.02
	3	292.68	292.03	0.22	292.68	292.03	0.22	292.68	292.03	0.22
	4	304.45	303.11	0.44	304.45	303.11	0.44	304.45	303.11	0.44

## Data Availability

Not applicable.

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
