# Peer review of "Effect of Multiple Annular Plates on Vibration Characteristics of Laminated Submarine-like Structures"

_materials, 2022, doi:10.3390/ma15186357_

Round 1
Reviewer 1 Report
The paper has been devoted to study the vibrational behavior of composite submarine-like shell structures. The structure consists of three section including conical, cylindrical and spherical shells. In addition, several annular plates have been used as internal sections. FSDT has been employed for formulating the shear deformation and Ritz method has been used as a numerical solution method. The main goal of the research is to obtain the natural frequency of the structure. The topic of the research is interesting and the novelty is sufficient. Moreover, the paper has been well-organized, however, there are some points which should be considered by the authors, as follows:
1- Some quantitative significant results should be mentioned in the abstract and conclusion.
2- Please explain that the application of the present structure in the specific submarine structure. It is better to present the name of application of that submarine structure.
3- The literature review must be improved. There are several studies in which the vibrational behavior joined shells and application of them have been investigated while the authors have not review them. Some of them are suggested as below:
https://doi.org/10.1016/j.ast.2021.107257, https://doi.org/10.1007/s00419-020-01715-1, https://doi.org/10.1016/j.ast.2021.107111 and so on.
4- In Table 2, the authors have discussed "first five mode" while the results of only four modes have been presented.
5- Figure 4 (a-d) is not clear. It seems that all parts of this figure are the same and there is no difference between the results depicted in these four parts. Please explain the main aim of presenting this figure.
6- The previous comment is also stated for the figure 5. Please check it.
7- The main original results which can be only obtained from this study should be presented in the conclusion. Moreover, it is better that some quantitative results are presented at the end of the conclusion.
Author Response
Thank you for your valuable suggestions, the details of our reply are given in accessory.

Reviewer 2 Report
The research is well designed. However, a few improvements should be introduced into the presentation.
Considering the Literature. All of the Authors of the papers should be mentioned, not only the first one. All of the co-authors work should be acknowledged.
Since the paper is about thin plates and shells, the appropriate citations should be added. Even though, they can look like textbooks now (for example the book by Flugge).
The use of abbreviations should be preceeded by the full name of the method (FEM, FSDT). There should be given a citation to basic texts on them.
There should be given more details about the finite element model, for example, a figure pointing out the discretization of the structure, the number of S4R elements, etc.
It would be good if a professional writer looked into the paper improving the style that seemes to be a bit heavy. I think the word "coupled" is used to often. It should be rather used "assembled structure". The word "coupled" fits rather to a set of equations, etc.
Author Response
Thank you for your careful reading and valuable suggestions, the details of our reply and corrections are given in the accessory.

Round 2
Reviewer 1 Report
Revisions have been done.
Author Response
Thanks for your valuable suggestions.
Reviewer 2 Report
The Authors introduced the amendments.
There is still a problem with the literature. Namely, the citations are not introduced correctly. The format of the literature positions has to be coherent and follow the rules of the Journal. In each record, the year, the numbers of pages and the volume number should be given. If relevant, the issue number should be given as well.
The literature section has to be corrected.
In my opinion, the paper can be published when the literature section will be improved.
Author Response
Thank you for your careful reading and valuable comments. According to your comment, the references are corrected in the correct form, including Ref.[1], Ref.[6], Ref.[7], Ref.[10], Ref.[13], Ref.[15], Ref.[17], Ref.[20], Ref.[22], Refs.[28~31], Ref.[35]. Please see the attachment.
